# Cellular stressors contribute to the expansion of hematopoietic clones of varying leukemic potential

Terrence N. Wong *et al.*[#]

Hematopoietic clones harboring specific mutations may expand over time. However, it remains unclear how different cellular stressors influence this expansion. Here we characterize clonal hematopoiesis after two different cellular stressors: cytotoxic therapy and hematopoietic transplantation. Cytotoxic therapy results in the expansion of clones carrying mutations in DNA damage response genes, including *TP53* and *PPM1D*. Analyses of sorted populations show that these clones are typically multilineage and myeloid-biased. Following autologous transplantation, most clones persist with stable chimerism. However, *DNMT3A* mutant clones often expand, while *PPM1D* mutant clones often decrease in size. To assess the leukemic potential of these expanded clones, we genotyped 134 t-AML/t-MDS samples. Mutations in non-*TP53* DNA damage response genes are infrequent in t-AML/t-MDS despite several being commonly identified after cytotoxic therapy. These data suggest that different hematopoietic stressors promote the expansion of distinct long-lived clones, carrying specific mutations, whose leukemic potential depends partially on the mutations they harbor.

Hematopoietic stem cells (HSCs) acquire somatic mutations with age, resulting in a genetically heterogeneous population with each HSC possessing its own set of unique mutations[1]. Some of these mutations confer a fitness advantage, allowing the HSCs harboring them to clonally expand. Indeed, expanded hematopoietic clones with specific mutations have been observed in healthy individuals without known hematologic abnormalities[2–4]. This has been termed clonal hematopoiesis of indeterminate potential[5], and these clones have typically been found to carry mutations in epigenetic modifiers such as *DNMT3A*. Hematopoietic clonal expansion has also been observed following certain cellular stressors including aplastic anemia[6] and cytotoxic therapy[7,8]. In these cases, the expanded hematopoietic clones often carried mutations in a different set of genes (i.e., *PIGA*, *BCOR*, and *BCORL1* in the former and *TP53* and *PPM1D* in the latter), suggesting that cellular stressors influence hematopoietic clonal evolution. However, controlled studies investigating the degree to which specific stressors influence clonal expansion and characterizing the nature of these expanding hematopoietic clones are lacking.

To address these questions, we use a sensitive error-corrected sequencing approach to characterize clonal hematopoiesis after two distinct types of hematopoietic stress: cytotoxic therapy and hematopoietic autologous transplantation. We find that the stress from cytotoxic therapy promotes the expansion of clones with mutations in DNA damage response genes such as *TP53* and *PPM1D*. Transplantation-related stress selects for a different set of hematopoietic clones, including those harboring *DNMT3A* mutations. Expanded clones are typically long-lived, multilineage, and transplantable. Finally, the leukemogenic potential of these clones is strongly dependent on the specific mutations they carry.

## Results

**Influence of cytotoxic therapy on clonal expansion.** To assess how cytotoxic therapy influences the expansion of specific hematopoietic clones, we analyzed mobilized pre-transplant pheresis samples from three age and gender-matched cohorts: 69 lymphoma patients, 50 myeloma patients, and 19 healthy donors (Table 1, Supplementary Data 1 and Supplementary Table 1). Of the 119 patients with a history of malignancy, 81 (68 lymphoma patients and 13 myeloma patients) had previously received cytotoxic therapy (i.e., chemotherapy and/or radiation) and were grouped together. We sequenced 46 genes associated with clonal hematopoiesis or AML/MDS (Supplementary Table 2) using an error-corrected next-generation sequencing assay capable of detecting somatic mutations at a variant allele frequency (VAF) of 0.1%. In total, 272 somatic nucleotide variants (SNVs) and 55 insertions/deletions (indels) were detected

(Supplementary Data 2). The median VAF of these variants was 0.5% (range: 0.1–22.6%) and similar across genes (Supplementary Fig. 1A). Over eighty-six percent had a VAF < 2%, the proposed threshold to detect clonal hematopoiesis[5]. Despite this lower VAF threshold, ninety percent of the identified SNVs were missense or nonsense mutations, and over 94% of the indels resulted in a frameshift (Supplementary Fig. 1B, 1C). This suggests that most of the identified variants had protein modifying activity.

Similar to earlier observations[7,8], 28.4% of patients exposed to cytotoxic therapy had clonal hematopoiesis carrying a mutation with a VAF ≥ 2%; however, 82.7% of these same patients had clonal expansion with a VAF ≥ 0.1% (Fig. 1a). Both the incidence of clonal hematopoiesis and the number of somatic variants were higher in patients receiving cytotoxic therapy versus patients with malignancy not receiving cytotoxic therapy or healthy donors (Fig. 1a, b, Supplementary Table 3). This was not dependent on the patient's initial diagnosis, suggesting that the increase in identified variants was primarily due to the treatment they received (Supplementary Fig. 2). Consistent with prior reports, the number of variants correlated with age[2–4,7,8], regardless of antecedent cytotoxic therapy (Fig. 1c). Indeed, variants detected after cytotoxic therapy were predominantly transition mutations (Fig. 1d) and occurred in a similar trinucleotide context as variants identified in the absence of such exposure (Supplementary Fig. 1D), suggesting that they are age-related, not induced by cytotoxic drugs. As in studies with healthy individuals[2–4], *DNMT3A* and *TET2* variants were common in all cohorts (Fig. 1e). In contrast, consistent with studies on patients with a history of malignancy[7,8], *TP53* and *PPM1D* variants were enriched after cytotoxic therapy (Supplementary Table 3) with all *PPM1D* variants being exon 6 nonsense or frameshift mutations (Supplementary Fig. 3).

Of the 46 genes interrogated, we identified six commonly associated with the DNA damage response pathway: *TP53*, *PPM1D*, *ATM*, *BRCC3*, *SRCAP*, and *RAD21*[9–12]. Whereas 38/81 (46.9%) of patients exposed to cytotoxic therapy had one or more variants in these genes, they were present in only 9/57 (15.8%; *P* = 0.0001 by Fisher's exact) of patients lacking such exposure (Fig. 1f). In contrast, variants in the other 40 genes were not significantly increased after cytotoxic therapy (Fig. 1g). Interestingly, 23/81 (28.4%) of patients exposed to cytotoxic therapy had a variant in a DNA damage response gene other than *TP53* or *PPM1D*. This was significantly higher than the 4/57 (7.0%; *P* = 0.002 by Fisher's exact) of individuals without such exposure and suggests that exposure to cytotoxic therapy may confer a fitness advantage to hematopoietic clones harboring mutations in multiple genes involved in DNA damage response, not just clones with *TP53* and *PPM1D* mutations.

In previous analyses of patients with hematologic and solid malignancies and concurrent clonal hematopoiesis, multiple

**Table 1 Clinical summary of patients with error-corrected sequencing**

|  |  | Lymphoma (*n* = 69) | Myeloma (*n* = 50) | Healthy donor (*n* = 19) |
|---|---|---|---|---|
| Age | Median (range) (years) | 57 (18–72) | 63 (26–76) | 60 (21–76) |
| Gender | Male | 60.9% | 60.0% | 63.2% |
|  | Female | 39.1% | 40.0% | 36.8% |
| Known previous treatment | Cytotoxic chemotherapy | 98.6% | 4.0% | N.A. |
|  | Alkylator | 95.7% | 0.0% | N.A. |
|  | Topoisomerase II inhibitor | 95.7% | 4.0% | N.A. |
|  | Platinum | 59.4% | 0.0% | N.A. |
|  | Radiation | 20.3% | 24.0% | N.A. |
|  | Proteasome inhibitor | 1.4% | 84.0% | N.A. |
|  | Lenalidamide/thalidomide | 0.0% | 84.0% | N.A. |
| Time from start of cytotoxic therapy to transplant | Median (range) (months) | 11 (3–288) | N.A. | N.A. |

N.A. not applicable

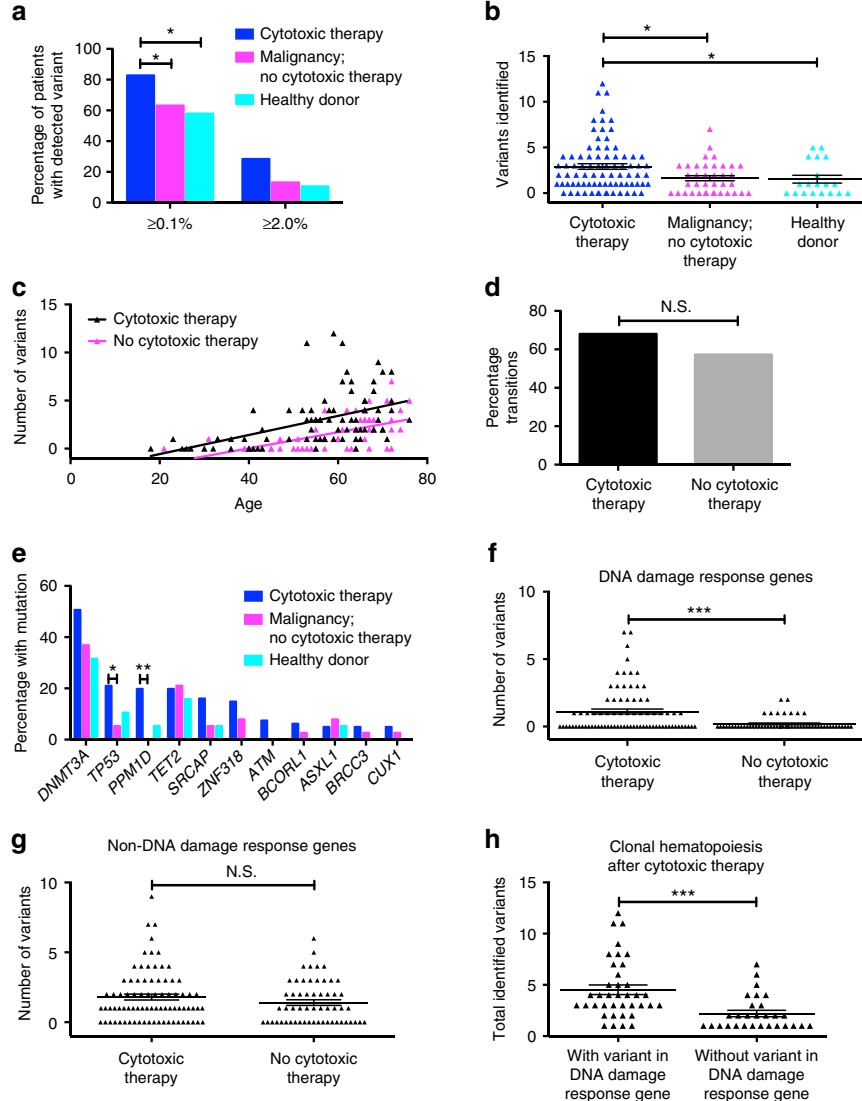

**Fig. 1** Influence of cytotoxic therapy on clonal expansion. **a** Percentage of individuals with clonal hematopoiesis ≥ 0.1% or ≥ 2.0% in three cohorts: those exposed to cytotoxic therapy (*n* = 81), those with malignancy but not exposed to cytotoxic therapy (*n* = 38), and healthy donors (*n* = 19). The average ages for the three groups were 55.0, 60.1, and 55.8 years, respectively. **b** Number of detected variants per patient. **c** Number of identified variants plotted against individual age for all pheresis samples. **d** Percentage of variants detected in pheresis samples that were transition mutations. **e** Percentage of individuals with at least one expanded clone harboring a variant in the specified gene. **f** Total number of variants identified in DNA damage response genes (i.e., *TP53*, *PPM1D*, *ATM*, *BRCC3*, *SRCAP*, or *RAD21*) grouped by whether or not the individual had previously been exposed to cytotoxic therapy. **g** Same as in **f** except for all other variants (i.e., in "non-DNA damage response genes"). **h** Total number of variants detected in patients with clonal hematopoiesis following cytotoxic therapy grouped by whether or not at least one variant was in a DNA damage response gene. NS not significant. \**P* < 0.05; \*\**P* < 0.01; \*\*\**P* < 0.001. For **a**, **d**, **e**, significance was determined with a Fisher's exact test. For **b**, significance was determined with a negative binomial regression analysis using Bonferroni-adjusted pairwise comparisons. For **f**–**h**, significance was determined with a negative-binomial regression analysis. Data is represented as the mean ± the standard error of the mean

variants were identified in 30.8 and 38% of patients, respectively[7,8]. With the increased sensitivity of our error-corrected sequencing approach, we identified multiple variants in 73.1% (49/67) of patients with clonal hematopoiesis after cytotoxic therapy (range: 2–12 variants), including 21 patients (31.3%) with multiple variants in DNA damage response genes (Fig. 1b, Supplementary Fig. 4). Of those individuals with multiple variants, 59.2% (29/49) had multiple variants in the same gene, with 13 patients (26.5%) having three or more (maximum: seven variants). As previously demonstrated with *PPM1D* alone[8], patients with at least one mutation in a DNA damage response gene after cytotoxic therapy had a higher

number of total variants than those with clonal hematopoiesis due to other mutations (Fig. 1h).

**Expanded clones are usually multi-lineage and myeloid-biased.** To investigate the nature of expanded mutant clones, we assessed the hematopoietic populations carrying their mutations. In pre-leukemic clones, the founding mutation is often in both myeloid and T cell compartments, suggesting that they arise in progenitors with multilineage potential[13–15]. We performed similar analyses with lymphoma patients after cytotoxic therapy. Myeloid and lymphoid populations were sorted from pheresis samples

with *DNMT3A*, *PPM1D*, or *TP53* mutations, and droplet digital PCR (ddPCR) was performed to quantify VAFs (Supplementary Fig. 5 and 6). Of note, B cells were often undetectable, reflecting recent rituximab exposure. As previously reported[16], *DNMT3A* mutations were detected in both myeloid and lymphoid cells, suggesting they arose in HSCs (Fig. 2a). With one exception (UPN-746), *PPM1D* and *TP53* mutations were also detected in both myeloid and lymphoid cells, suggesting that they likewise arose in HSCs (Fig. 2b, c). Interestingly, the percentage of cells carrying the *TP53* or *PPM1D* variant was higher (median: 14.2-fold; range: 2.3 to 88.3-fold; $P = 0.03$ by Wilcoxon match-paired signed rank test) in myeloid versus T cells. This suggests that the *TP53* and *PPM1D* variants carried by hematopoietic clones expanding after cytotoxic therapy are initially and rapidly propagated through the myeloid lineages. Either the progenitors harboring these variants were myeloid-biased or didn't have time to fully populate the lymphoid lineages, particularly T cells, which are slower to turn over. More than half of our samples were collected within 1 year of cytotoxic therapy initiation, including UPN-746 (Fig. 2, Supplementary Data 1).

**Influence of transplantation on clonal expansion**. We then asked how transplantation influences the expansion of these HSCs. Transplantation causes replicative stress[17,18] and alters the bone marrow microenvironment[19–21], potentially influencing the competitive fitness of HSCs. Forty lymphoma patients (with 120 detected pheresis variants among 31 individuals) had peripheral blood samples collected 6–12 months after autologous transplantation. Most of these variants did not change significantly in VAF upon transplantation with their post-transplant VAFs strongly correlated to their VAFs before transplantation ($R^2 = 0.64$ by linear regression); however, variants not initially identified in the pheresis samples often became detectable post-transplant (Fig. 3a, b, Supplementary Data 3). In total, 148 variants were called in 35 patients following autologous transplantation with the average number of variants per patient increasing slightly from $3.0 \pm 3.1$ to $3.7 \pm 3.2$ ($P < 0.05$ by Wilcoxon match-paired signed rank test).

The two genes with the most variants identified either before or after transplant were *DNMT3A* and *PPM1D*. Of 51 *DNMT3A* variants, 17 (33.3%) significantly increased ≥ 2-fold in VAF after transplantation, while 3 (5.9%) decreased (Fig. 3c, e). Interestingly, the VAFs of all three R882 codon *DNMT3A* variants increased. This suggests that certain *DNMT3A* variants confer a modest repopulating advantage with transplantation. In contrast, of 23 *PPM1D* variants, only 2 (8.7%) significantly increased in VAF with transplantation, while 7 (30.4%) decreased (Fig. 3d, e). Collectively, these data indicate that hematopoietic clones expanding after cytotoxic therapy are often long-lived, persisting following transplant. At the same time, previously unidentified clones may expand under the stress of transplantation. The behavior of hematopoietic clones following transplantation depends, in part, on what mutations they harbor; transplantation-induced selection pressure doesn't necessarily favor the same clones as cytotoxic therapy.

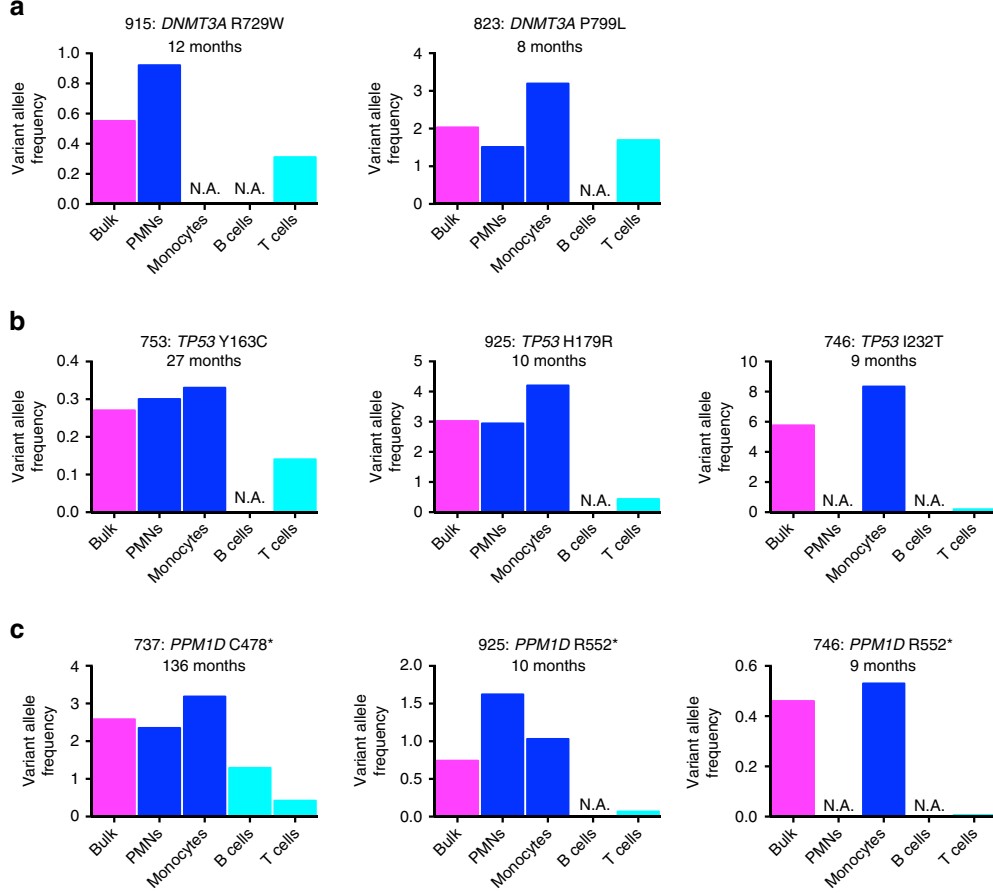

**Fig. 2** Expanded clones are usually multi-lineage and myeloid-biased. VAFs of the specified *DNMT3A* (**a**), *TP53* (**b**) or *PPM1D* (**c**) mutations in bulk cells (magenta) or in sorted neutrophils (PMNs, blue), monocytes (blue), B cells (turquoise), or T cells (turquoise). N.A.: the hematopoietic population was not able to be obtained. The latency from the time of first cytotoxic therapy exposure to autologous stem cell transplant is noted

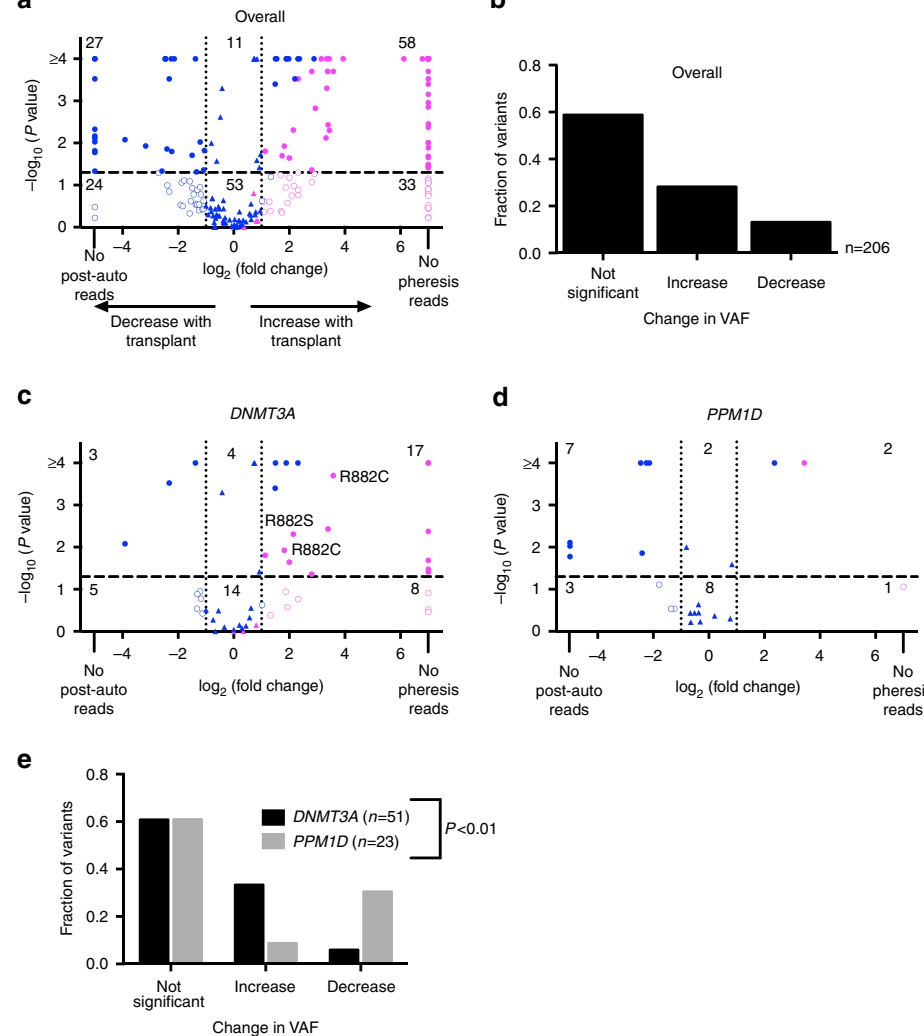

**Fig. 3** Influence of transplantation on clonal expansion. **a** Volcano plot showing the fold change in VAF following transplantation for those variants initially identified in pre-transplant pheresis samples (in blue) and those only detected after transplant (in magenta). Dotted and dashed lines respectively represent a greater than two-fold change in VAF and $P < 0.05$ by Fisher's exact or $\chi^2$ tests. Triangles indicate a less than two-fold change in VAF, while circles indicate a greater than two-fold change. Variants with open circles had a greater than two-fold change but with $P > 0.05$. **b**, Summary of the changes in VAF after transplantation for all genes; changes in VAF were deemed significant if they were > 2-fold with a $P$ value < 0.05. **c**, **d** Same as **a** except focusing on *DNMT3A* (**c**) or *PPM1D* (**d**) variants. **e** Same as **b** except summarizing the changes in VAF for *DNMT3A* and *PPM1D* variants. Significance for the difference in behavior after transplantation between variants in *DNMT3A* and *PPM1D* was determined with a $\chi^2$ analysis

**Leukemogenic potential of expanded clones**. Clonal hematopoietic expansion has been linked to the development of hematologic malignancies both with and without previous cytotoxic therapy exposure[2,3,8,22,23]. However, it remains unclear which clones are at highest risk of leukemic transformation. At a median follow-up of ~2.2 years, no patient in our lymphoma cohort had developed t-AML/t-MDS, so directly assessing the leukemogenic potential of expanded clones following cytotoxic therapy was not possible. Instead, we indirectly assessed this potential by determining how frequently these genes are mutated in t-AML/t-MDS. We performed exome sequencing on 19 cases and targeted sequencing on 92 cases with a panel of 274 genes, including the DNA damage-associated genes assessed in our error-corrected HaloPlex assay (Supplementary Table 4). Combined with our previously published t-AML/t-MDS whole genome and exome sequencing[24], we analyzed 134 cases in total (Table 2, Supplementary Data 4).

As in previous reports[24,25], these patients had poor overall survival (Supplementary Fig. 7). They frequently harbored

mutations in epigenetic modifiers, spliceosome genes, and myeloid transcription factors and were enriched in *TP53* mutations (Fig. 4a, Supplementary Data 5). The null hypothesis that all gene mutations confer a similar risk of progression from clonal hematopoiesis to t-AML/t-MDS predicts that the frequency of gene mutations in therapy-related clonal hematopoiesis will be similar to that in t-AML/t-MDS. However, this pattern was not observed as illustrated by DNA damage response genes (Fig. 4b). *TP53* mutations were frequently observed in both therapy-related clonal hematopoiesis and in t-AML/t-MDS. In contrast, mutations in *PPM1D* and *SRCAP*, which were present at approximately the same frequency as *TP53* mutations in therapy-related clonal hematopoiesis, were uncommon in t-AML/t-MDS. Similar trends were observed upon restricting t-AML/t-MDS patients to those with a previous history of lymphoma (Fig. 4c). This suggests that the leukemogenic potential of expanded clones carrying different mutations varies significantly. Specifically, the leukemogenic potential of expanded *TP53* mutant hematopoietic

**Table 2 Clinical summary of t-AML/t-MDS patients**

| | T-AML/t-MDS sequencing cohort | Overall (n = 134) | Exome (n = 20) | Targeted (n = 92) |
|---|---|---|---|---|
| Age | Median (range) (years) | 62.0 (18–85) | 61.5 (24–80) | 63.0 (18–85) |
| Gender | Male | 44.0% | 40.0% | 46.7% |
| | Female | 56.0% | 60.0% | 53.3% |
| Prior disease | Non-Hodgkin lymphoma | 29.9% | 50.0% | 29.3% |
| | Breast | 27.6% | 20.0% | 25.0% |
| | Gastrointestinal | 6.0% | 5.0% | 7.6% |
| | Hodgkin disease | 5.2% | 5.0% | 6.5% |
| | Prostate | 5.2% | 0.0% | 6.5% |
| | Testicular | 4.5% | 0.0% | 5.4% |
| | Other | 21.6% | 20.0% | 19.6% |
| Known previous treatment | Alkylator | 57.5% | 65.0% | 58.7% |
| | Topoisomerase II inhibitor | 50.7% | 45.0% | 48.9% |
| | Platinum | 20.9% | 20.0% | 21.7% |
| | Radiation | 65.7% | 65.0% | 65.2% |
| | Unknown | 8.2% | 15.0% | 5.4% |
| Latency | Median (range) (years) | 6.1 (0.7–31) | 6.8 (2.5–15.9) | 6.2 (0.7–31) |
| Diagnosis | AML | 50.0% | 40.0% | 40.2% |
| | MDS | 50.0% | 60.0% | 59.8% |
| Cytogenetics | Deletion 5 | 29.9% | 55.0% | 30.4% |
| | Deletion 7 | 33.6% | 70.0% | 32.6% |
| | Complex | 41.0% | 70.0% | 39.1% |
| | MLL abnormality | 10.4% | 5.0% | 8.7% |
| | Other/unknown | 38.8% | 0.0% | 41.3% |
| % blasts in the bone marrow | Median (range) (%) | 14 (0–95) | 8 (1–80) | 9 (0–91) |
| Most intensive treatment regimen | Allogeneic transplant | 37.3% | 30.0% | 41.3% |
| | Myeloablative | 21.6% | 30.0% | 15.2% |
| | Non-myeloablative | 30.6% | 30.0% | 34.8% |
| | Other/unknown | 10.4% | 10.0% | 8.7% |
| Remission | Yes | 49.3% | 30.0% | 53.3% |
| | No | 48.5% | 70.0% | 45.7% |
| | Unknown | 2.2% | 0.0% | 1.1% |
| Overall survival | Median (range) (days) | 385 (6–3427) | 187.5 (32–3427) | 424 (6–2432) |

histones into the nucleosome[26,27]. Interestingly, this protein has been shown to play a role in the repair of double-strand breaks resulting from genotoxic stress[9]. The 29 *SRCAP* variants identified in pheresis and/or post-transplant peripheral blood samples were scattered throughout the gene with 12 (41.4%) truncating nonsense or frameshift mutations and 16 (55.2%) missense mutations, suggesting they may be primarily loss-of-function. Of note, since we did not survey all DNA damage response genes, we may be underestimating the frequency of clonal hematopoiesis involving mutations in such genes following cytotoxic therapy. Indeed, Coombs et al. also identified mutations in the *TP53* regulator *CHEK2* in patients with solid malignancies[7].

In most patients with clonal hematopoiesis following cytotoxic therapy, we identified multiple variants. This has been observed to a lesser degree in other studies of clonal hematopoiesis after cytotoxic therapy[7,8] and in healthy individuals[2–4]. Determining whether these variants represent independent clones would require single-cell sequencing. However, the number of variants identified in individual patients (with many located in the same gene) and the wide distribution of their VAFs (Supplemental Data 2) suggest that at least some arise from independent clones. Consistent with this conclusion, Gibson et al. described patients with multiple variants after cytotoxic therapy who later developed t-AML/t-MDS without all the variants present in the malignant clone[8]. In patients with prior exposure to cytotoxic therapy, the presence of a mutation in a DNA damage response gene above the limit of detection was associated with a higher total number of identified variants. This observation raises the possibility that the expansion of hematopoietic clones harboring mutations in *TP53, PPM1D,* and other DNA damage response genes may serve as a biomarker of prior genotoxic stress. Several studies have shown that clonal hematopoiesis in patients following cytotoxic therapy is associated with a modestly increased risk of developing a therapy-related myeloid neoplasm[8,22,23]. Whether the presence and/or number of expanded clones with mutations in DNA damage response genes better defines the risk for therapy-related myeloid neoplasms will require further study.

In studies involving individuals with no known hematologic malignancy or with aplastic anemia, expanded hematopoietic clones have often been found to persist for years, often at stable VAFs, suggesting that they originated in an HSC or long-term progenitor[3,6,16]. Indeed, we show that mutant clones that have expanded following cytotoxic therapy typically exhibit multi-lineage potential, expanding initially through the myeloid lineages. These clones also were typically long-lived, persisting following autologous transplantation. In total, these data suggest that hematopoietic clones expanding under the stress of cytotoxic therapy typically arise from HSCs.

Our data suggests that different cellular stressors favor distinct hematopoietic clones harboring specific mutations and is consistent with murine models of how mutant HSCs respond to cellular stress. HSCs with loss of the DNA damage response gene *Trp53* gain a fitness advantage with genotoxic stress potentially through p53-mediated cell competition[24,28]. However, transplantation-induced stress does not provide them with a significant competitive advantage[24,28–30]. In contrast, *Dnmt3a* deficient HSCs gain a repopulating advantage over their wild-type counterparts with serial transplantation[31]. In humans, expanded *DNMT3A* mutant clones have been identified in aged individuals[2–4]. They have also been observed actively expanding in the setting of aplastic anemia, where they portend a poor prognosis[6]. The cellular stressors influencing *DNMT3A* mutant clonal expansion in these two cases and how they relate to transplantation-induced stress remain unknown.

Our findings are also consistent with observations of patients undergoing hematopoietic transplantation. In patients receiving

clones appears to be higher than clones with mutations in other DNA damage response genes, particularly *PPM1D* and *SRCAP*.

## Discussion

In this study, we show that hematopoietic clones harboring mutations in DNA damage response genes expand in response to cytotoxic stress. The expansion of *TP53* and *PPM1D* mutant clones has previously been observed after cytotoxic therapy in patients with either solid or hematologic malignancies[7,8]. Our data suggest that mutations in DNA damage response genes other than *TP53* and *PPM1D* may also provide a fitness advantage to HSCs with cytotoxic therapy. In the earlier studies, mutations in some of these genes (e.g., *ATM*, *BRCC3*, etc.) were identified in a small percentage of patients[7,8]. Utilizing the sensitivity of error-corrected sequencing, we show that clonal hematopoiesis involving such mutations is more frequent after cytotoxic therapy than previously observed. We also ascertain for the first time that mutations in *SRCAP* are frequently seen in patients following cytotoxic therapy. *SRCAP* is an ATP-dependent chromatin remodeler, which catalyzes the incorporation of variant H2A.Z

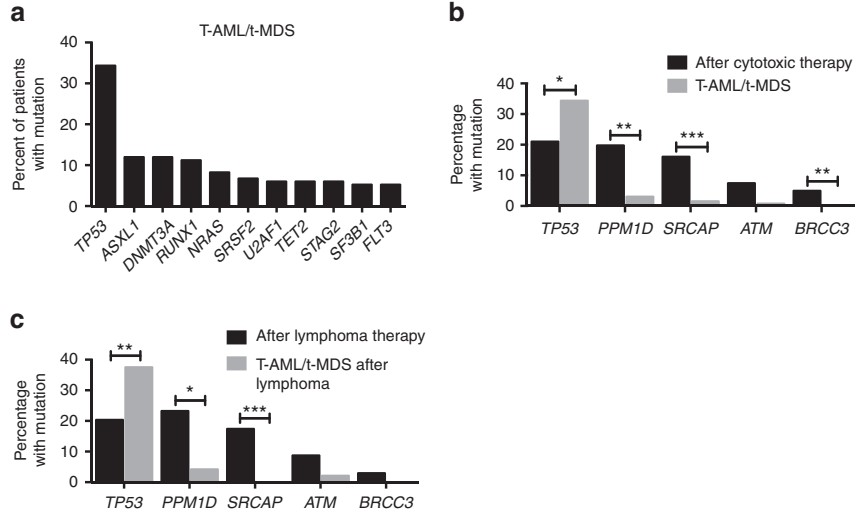

**Fig. 4** Leukemogenic potential of expanded clones. **a** Percentage of t-AML/t-MDS cases ($n = 134$) harboring a mutation in the indicated gene. **b** Percentage of patients who after cytotoxic therapy (n=81) had an expanded clone harboring a variant in the indicated DNA damage response gene compared to the percentage of t-AML/t-MDS patients ($n = 134$) with a mutation in the same gene. **c** Same as **b** except that the comparison is between lymphoma patients at the time of autologous transplant ($n = 69$) and t-AML/t-MDS patients with a primary diagnosis of lymphoma ($n = 48$). For **b**, **c** significance was determined with an exact logistic regression through use of a joint test. *$P < 0.05$; **$P < 0.01$; ***$P < 0.001$

an autologous transplant, the risk of t-AML/t-MDS is related more to the extent of pre-transplant cytoreductive therapy than transplantation itself[32–34], potentially because transplantation-induced selection pressure does not provide a further fitness advantage to the mutant hematopoietic clones expanding with cytotoxic therapy. In the case of allogeneic transplantation, donor-derived *DNMT3A* mutant hematopoietic clones have been identified both in recipients with unexplained cytopenias after transplant and in recipients developing donor cell leukemia[35–38]. The role of screening potential allogeneic donors, particularly those of advanced age, for clonal hematopoiesis will be an area of future research interest, particularly given the transplantability of certain mutant clones.

Finally, our sequencing of t-AML/t-MDS suggests that the leukemogenic potential of expanded hematopoietic clones is strongly dependent on the specific mutations they harbor with *TP53* mutant clones having a higher propensity towards leukemic evolution than clones with mutations in other DNA damage response genes, including *PPM1D*. In contrast to our findings, Lindsley et al. identified a high frequency (15%) of *PPM1D* mutations in t-MDS[39]. A potential explanation for this discrepancy is the possibility that long-lived expanded non-leukemic clones may co-exist with leukemic clones (Supplementary Fig. 8). Supporting this is the observation that non-leukemic clones harboring mutations in *TP53*, *DNMT3A*, *PPM1D*, *SRCAP* and *ZNF318* may rapidly expand with treatment of the AML/MDS clone[40–42]. In the Lindsley et al. study, the median VAF of the largest *PPM1D* clone was only 5.5% compared to 14.5% for *TP53*, with over half the patients analyzed after receiving treatment for their MDS[39]. Thus, some *PPM1D* mutations may be unrelated to the founding MDS clone and instead represent co-existing non-leukemic clonal hematopoiesis. Despite being distinct from the leukemic clone, non-leukemic expanded clones may still inform on its evolution (e.g., suggest a previous exposure to genotoxic stress) and provide prognostic information.

In summary, our data show that cellular stressors result in the expansion of hematopoietic clones carrying specific mutations. Genotoxic stress from cytotoxic therapy promotes the expansion of clones with mutations in DNA damage response genes, such as *TP53* and *PPM1D*. Transplantation-related stress may select for

clones with *DNMT3A* mutations, particularly in codon 882. In the future, longitudinal studies of patients before and after specific stressors (e.g., cytotoxic therapy, mobilization ± cytotoxic stimulation, transplantation, etc.) will more precisely define their influence on clonal expansion and distinguish the impact of these stressors from that of other factors (e.g., pre-existing malignancies). Expanded clones are often long-lived and transplantable, but their leukemogenic potential varies, in part, due to the mutations they harbor. Most expanded clones, even those harboring *TP53* mutations, do not evolve into leukemia. The contribution of other cellular stressors, such as inflammation, to the development of clonal hematopoiesis and the role that cell-extrinsic stressors and/or cell-intrinsic genetic (or epigenetic) alterations play in the progression from clonal hematopoiesis to leukemia remain open and important questions.

## Methods

**Patient characteristics**. Lymphoma patients undergoing autologous stem cell transplantation with available pheresis samples were consented under IRB protocol #201108251. Lymphoma patients were analyzed based on sample availability. Myeloma patients undergoing an autologous stem cell transplant and allogeneic stem cell transplant donors with available pheresis samples were consented under IRB protocol #201102270. Myeloma patients and healthy donors were chosen to allow for age and gender-matching with the lymphoma cohort. No data were excluded from the analysis. T-AML/t-MDS patients were selected from a larger cohort of adult AML and MDS patients enrolled in a single institution tissue banking protocol that was approved by the Washington University Human Studies Committee (IRB protocol #201011766). Written informed consent for sequencing, including whole-genome sequencing, was obtained from all study participants. Patients were treated in accordance with National Comprehensive Cancer Network guidelines (http://www.nccn.org) with an emphasis on enrollment in therapeutic clinical trials whenever possible.

Patients were considered to have a previous history of cytotoxic therapy exposure if they previously received an alkylator, topoisomerase II inhibitor, or platinum-based chemotherapy or radiation therapy. Cytotoxic agents given as part of the pheresis mobilization regimen were not considered as a prior exposure to cytotoxic therapy given the short time period between this exposure and pheresis collection. Overall, 81 patients with analyzed pheresis samples had prior exposure to cytotoxic therapy, and 57 individuals lacked such exposure. To detect a change in the incidence of clonal hematopoiesis ≥ 0.1% VAF from 50% without cytotoxic therapy to 75% after cytotoxic therapy exposure with 80% power at 0.05 significance, we had calculated a desired sample size of 58 individuals per cohort. Clinical data for all patients in this study are included in Tables 1 and 2, Supplementary Table 1, and Supplementary Datas 1 and 4.

**Error-corrected HaloPlex sequencing.** Error-corrected sequencing was performed using the HaloPlex HS Target Enrichment System (Agilent Technologies) as previously described[43] with modifications as noted below. Briefly, as per Version C1 of the manufacturer's protocol, up to 500 ng of genomic DNA was digested with a custom restriction enzyme cocktail in eight separate reactions and subsequently hybridized to a customized HaloPlex HS probe library × 2 hours. This library was designed using the Agilent SureDesign platform to target 46 genes (Supplementary Table 2). The probes had dual indices: a unique molecular barcode to allow for error-corrected sequencing and a sample index to allow for sample multiplexing. Hybridized genomic DNA fragments were subsequently ligated to the library probes, captured with streptavidin, and amplified with PCR (× 24 cycles), creating read families, each with its own unique molecular barcode index. Library quality was assessed with the Agilent 2100 Bioanalyzer. Library concentration was assessed with qPCR according to the manufacturer's protocol (Kapa Biosystems). Libraries were normalized, pooled, and sequenced on the HiSeq 4000 with 15 samples sequenced per lane. On average, each sample had 40.7 million ± 13.0 million mapped reads.

Molecularly barcoded reads were aligned with BWA MEM v0.7.9a, using parameters "-t 12, -M". SNVs were detected using BarCrawler, a custom GATK walker (https://github.com/abelhj/gatk/blob/master/public/external- example/src/main/java/org/abelhj/WalkerTR1203.java), using parameters "-mmq 20 -mbq 20- minCtBC 3 -dcov 500000 -discardN 1 -minOffset 5 -maxNM 3". Filters were applied to remove artifacts appearing at homopolymer runs of length greater than 4 and alignment artifacts appearing in greater than 5% of a panel of normal samples. Next, background noise calculation was performed on a position-by-position basis for each identified variant as follows. For each variant, read counts were gathered from all other samples, excluding those sites with a VAF above 25%, which were assumed to be germline single nucleotide polymorphisms. The R statistical package was used to obtain a *p*-value via Fisher's exact test, comparing the reference and variant reads at a site to the number of reference and variant reads at that site in all other samples. Multiple testing correction was then applied with the *p* adjust function (default parameters). Those variants with an adjusted *p*-value of less than 0.1 were retained. The same process was then repeated with subsequent background calculations excluding all variants retained in previous rounds until no new variants were identified. SNVs were retained if they were coding or splice site mutations, had 500 or more total reads, had 4 or more supporting reads, had a VAF greater than or equal to 0.1%, and were not present in the Exome Aggregation Consortium database at a frequency greater than 0.1%. Finally, three variants (*CREBBP* L2067, *SETBP1* L284, and *ZNF318* A1466S), which were present at varying frequencies in most samples tested, were excluded as artifacts.

Insertions and deletions were detected by using BarCrawler Consensus (https://github.com/abelhj/gatk/blob/master/public/external-example/src/main/java/org/abelhj/WalkerTRConsensus.java) to assemble each read family into a consensus sequence (params: -dcov 500000). Then VarScan v2.3.6 (params --min-coverage 3 --min-reads 3 --min-var-freq 0.001) was run on the resulting bam file to identify putative indels. As with SNVs, variant calls were restricted to coding regions of the genome. A per-site background error rate calculation was based on the total counts of any indel appearing at each position within each batch, thus accounting for artifacts like slippage at homopolymer runs. A Fisher's exact test was used to compare each variant call to the background rate; then p-values were corrected for multiple testing using the Bonferroni method. Indels with adjusted *p* < 0.01 were retained. This process was applied iteratively, as with the SNVs above. To account for highly variable sites, variants were also excluded if their VAF did not exceed five times the median absolute deviation of all VAFs at that location. Review of known problematic positions like *ASXL1* G545 show that despite high levels of background "noise", obvious sequencing artifacts were removed. Both SNVs and indels were identified without a priori knowledge of whether the sample of interest was a pheresis or post-transplant peripheral blood sample or whether the pheresis sample was banked after a previous exposure to cytotoxic therapy.

**Statistical analyses.** In determining the relationship between hematopoietic clonal expansion and an individual's clinical features (Supplementary Table 3), the outcomes of interest included the identification of clonal hematopoiesis with a VAF ≥ 0.1%, the total number of variants identified, and the presence of a variant in the genes *DNMT3A*, *PPM1D*, *TP53*, *SRCAP*, *TET2*, *ZNF318*, or *ATM*. The covariates included age and a history of radiation therapy, chemotherapy, or cytoreductive therapy (i.e., either radiation therapy or chemotherapy). The relationships between the binary outcomes and the covariates were analyzed with exact logistic regression. This was done in preference to ordinary logistic regression due to the small sample sizes available for some combinations of outcomes and covariates. The relationship between each outcome and each covariate, both alone and in combination with age, was examined. The relationships between the number of mutations and the covariates, both alone and in combination with age, were analyzed with a negative-binomial regression analysis. This technique is not dependent on equal variances between groups and is appropriate when the outcome is a count variable.

The question of whether mutations in the DNA damage response genes *TP53*, *PPM1D*, *SRCAP*, *BRCC3*, and *ATM* had a similar distribution after cytotoxic therapy versus at t-AML/t-MDS was examined with an exact logistic regression through use of a joint test of the hypothesis that their parameters are equal to zero.

The analyses were done with the logistics procedure of SAS/Stat software, Version 9.3 for Windows.

**Cell sorting.** Cryopreserved cells were thawed as previously described[44]. $1.5 \times 10^7$ cells were stained by standard protocols with the following antibodies: anti-CD3 e450 (eBioscience; clone OKT3; catalog# 48-0037-42; 55:1 dilution), anti-CD14 APC (eBioscience; clone 61D3; catalog# 17-0149-42; 20:1 dilution), anti-CD15 FITC (Fisher; clone HI98; catalog# BDB555401; 4.6:1 dilution), anti-CD16 PE (Fisher; clone 3G8; catalog# BDB555407; 11:1 dilution), and anti-CD19 APC (Fisher, clone HIB19; catalog# BDB555415; 14:1 dilution). They were sorted using a modified Beckman Coulter MoFlo. Genomic DNA was isolated with the QIAmp DNA Mini Kit (Qiagen, Venlo, The Netherlands).

**Droplet digital PCR.** All primers and probes for ddPCR were designed by Bio-Rad per MIQE guidelines. ddPCR was performed as previously described[45]. Specifically, PCR was performed with 900 nM forward and reverse primers, 250 nM mutant and wild-type probes, and < 2 ng/μl genomic DNA with restriction enzyme digestion per Bio-Rad recommendations. PCR was performed with annealing/extension temperatures of 53.5–56.0 °C for 40 cycles. For droplet generation and analysis, we used the Bio-Rad QX200™ Droplet Digital™ PCR System. Calculation of the mutant allele fraction was performed as previously described[24]. Specifically, we only identified mutant alleles when present in droplets also lacking the reference allele. We then used Poisson statistics to determine the number of droplets containing a single allele (either reference or mutant) and the number of droplets containing a single mutant allele. The VAF was determined as the percentage of the single allele droplets containing the mutant allele. Using "mutant only" droplets to determine the mutant allele fraction significantly reduces artifact caused by DNA degradation that has resulted in the chemical alteration of one of the two DNA strands (i.e., guanosine oxidation, cytosine deamination, etc.).

**Targeted library preparation.** Automated dual-indexed libraries were constructed with 250 ng of genomic DNA utilizing the KAPA HTP library prep kit (KAPA Biosystems) on the SciClone NGS platform (Perkin Elmer) for 96 tumor/normal pairs. The samples were fragmented on the Covaris LE220 (Covaris) targeting 250 bp inserts. Forty-eight of the libraries were pooled pre-capture generating a 5 μg library pool. Each library pool was hybridized with a custom set of xGen Lockdown Probes (IDT) targeting a set of recurrently mutated genes (RMG) found in AML. An additional ten commonly identified by our error-corrected assay as mutated after cytotoxic therapy were added at an equal molar ratio to the RMG probes, thus representing all regions with a similar quantity (Supplementary Table 4). The libraries were hybridized for 4 hours at 47 °C followed by stringent washing. Enriched ssDNA library fragments were amplified with KAPA HiFi HotStart polymerase and 200 nM primers prior to sequencing. The concentration of each captured library pool was determined through qPCR according to the manufacturer's protocol (KAPA Biosystems) to produce cluster counts appropriate for the Illumina HiSeq4000 platform. The capture pools were normalized and pooled prior to data generation. One lane of 2 × 150 bp sequence data generated ~1 Gb of data per sample. Approximately 300× mean depth of coverage was achieved, and 99.9% of targets were covered to a minimum of 20×.

**Exome library preparation.** Automated dual-indexed libraries were constructed with 250 ng of genomic DNA utilizing the KAPA HTP library prep kit (KAPA Biosystems) on the SciClone NGS platform (Perkin Elmer) for 19 tumor/normal pairs. The samples were fragmented on the Covaris LE220 (Covaris) targeting 250 bp inserts. Ten libraries were pooled pre-capture generating a 5 μg library pool. Each library pool was hybridized with the SeqCap EZ Human Exome Kit v3.0 (Roche Nimblegen) that targets over 200,000 genes spanning ~64 Mb of the human genome. In addition to the exome space, a custom set of xGen Lockdown Probes (IDT) was spiked into the hybridization reaction targeting a set of RMG found in AML. The additional probes were added at an equal molar equivalent to the probe concentration of the exome probes, thus representing the RMG regions in a similar quantity as the exome regions. The libraries were hybridized for 72 hours at 47 °C followed by stringent washing. Enriched ssDNA library fragments were amplified with KAPA HiFi HotStart polymerase and 200 nM primers prior to sequencing. The concentration of each captured library pool was accurately determined through qPCR according to the manufacturer's protocol (KAPA Biosystems) to produce cluster counts appropriate for the Illumina HiSeq2000 platform. Two lanes of 2 × 100 bp sequence data generated ~7 Gb of data per sample. Approximately 55× mean depth of coverage was achieved, and 83% of targets were covered to a minimum of 20×.

**Somatic variant analysis.** Sequence data was aligned to reference sequence build GRCh37-lite-build37 using bwa mem[46] version 0.7.10 (params: -t 8::) and then merged and deduplicated using picard version 1.113 (https://broadinstitute.github.io/picard/). SNVs were detected using the union of four callers: 1) samtools[47] version r982 (params: mpileup -BuDS) intersected with Somatic Sniper[48] version 1.0.4 (params: -F vcf −G -L -q 1 -Q 15) and processed through false-positive filter v1 (params: --bam-readcount- version 0.4 --bam-readcount-min-base-quality 15 --min-mapping-quality 40 --min-somatic-score 40), 2) VarScan[49] version 2.3.6

filtered by varscan-high-confidence filter version v1 and processed through false-positive filter v1 (params: --bam-readcount-version 0.4 --bam-readcount-min-base-quality 15), 3) Strelka[50] version 1.0.11 (params: isSkipDepthFilters = 1), and 4) mutect[51] v1.1.4 (params: number-of-chunks = 50). Indels were detected using the union of 4 callers: 1) GATK[52] somatic-indel version 5336 2) pindel[53] version 0.5 filtered with pindel somatic calls and VAF filters (params: --variant-freq-cutoff = 0.08), and pindel read support, 3) VarScan[49] version 2.3.6 filtered by varscan-high-confidence-indel version v1 and 4) Strelka[50] version 1.0.11 (params: isSkip-DepthFilters = 1). SNVs and indels were further filtered by removing artifacts found in a panel of 905 normal exomes, removing sites that exceeded 0.1% frequency in the 1000 genomes or NHLBI exome sequencing projects, and then using a Bayesian classifier (https://github.com/genome/genome/blob/master/lib/perl/Genome/Model/Tools/Validation/IdentifyOutliers.pm), retaining variants classified as somatic with a binomial log-likelihood of at least 10. Variants present at low VAF in the 46 genes interrogated by our HaloPlex error-corrected sequencing assay were manually reviewed if they had a binomial log-likelihood of at least three and retained if they passed validation.

**Data availability**. Exome sequencing data of t-AML/t-MDS has been submitted to dbGAP (accession number phs000159.v9). All other relevant data are included in the article or supplementary files, or available from the authors upon request. Code used for both error-corrected and non-error-corrected variant identification are described in their applicable methodology sections with download sites listed where applicable.

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

## Acknowledgements

This work was supported by National Institutes of Health, National Cancer Institute grants PO1 CA101937, (D.C.L. and T.J.L.), P50 CA171963 (D.C.L.) and K08 CA197369 (T.N.W.). Technical support was provided by the Alvin J. Siteman Cancer Center Tissue Procurement Core and the High-Speed Cell Sorting Core at Washington University School of Medicine, which are supported by National Cancer Institute grant P30 CA91842.

## Author contributions

T.N.W. designed and performed the research, analyzed the data, and wrote the manuscript. C.A.M., M.R.M.J., A.P.S., E.J.D. and R.S.F. contributed to the generation and analysis of the HaloPlex error-corrected, exome, and targeted sequencing data. N.B., A.F.C., N.M.H., M.F., S.E.H., M.J., K.L., P.W. and R.V. collected and processed clinical data and tissue samples. J.D.B. performed statistical analyses of the clinical data. J.F.D., J.S.W., T.A.G., M.J.W. and T.J.L. contributed to data analysis. D.C.L. supervised all the research and edited the manuscript, which was approved by all co-authors.

## Additional information

**Competing interests:** The authors declare no competing financial interests.

Terrence N. Wong[1], Christopher A. Miller[1,2], Matthew R.M. Jotte[1], Nusayba Bagegni[1], Jack D. Baty[3], Amy P. Schmidt[1], Amanda F. Cashen[1,4], Eric J. Duncavage[5], Nichole M. Helton[1], Mark Fiala[1], Robert S. Fulton[2], Sharon E. Heath[1], Megan Janke[1], Kierstin Luber[1], Peter Westervelt[1,4], Ravi Vij[1,4], John F. DiPersio[1,4], John S. Welch[1,4], Timothy A. Graubert[6], Matthew J. Walter[1,4], Timothy J. Ley[1,4] & Daniel C. Link[1,4]

[1]Division of Oncology, Washington University School of Medicine, St. Louis, MO 63110, USA. [2]McDonnell Genome Institute, Washington University School of Medicine, St. Louis, MO 63110, USA. [3]Division of Biostatistics, Washington University, St. Louis, MO 63110, USA. [4]Siteman Cancer Center, Washington University, St. Louis, MO 63110, USA. [5]Department of Pathology and Immunology, Washington University School of Medicine, St Louis, MO 63110, USA. [6]Massachusetts General Hospital, Boston, MA 02114, USA

