## [Peer Review File · Nature Communications]

Reviewers' Comments:

Reviewer #1:

Remarks to the Author:

In their revised manuscript, authors have addressed majority of the concerns raised by this and other reviewers. Despite authors' rebuts, the major findings in this manuscript have previously been suggested or easily anticipated. Nevertheless, this reviewer believes that it still merits the community, given the growing attention to age-related clonal hematopoiesis of recent years. There are a few additional minor points, in response to this revision.

1) Authors did not include conditioning regimens as "prior cytotoxic therapy exposure" since the time between such exposure and sample collection was typically only a few days. However, usually, it was not only a few days. It should be after a recovery from transient bone marrow suppression, which should surely not be just a few days. During conditioning, it could be possible that clonal selection does occur, provably promoting outgrowth of chemo resistant clones, which might be worth interrogating.

2) "Single-colony" sequencing would be accepted as a widely-used, reliable technique to address the origin of mutations in different population in well-controlled experiments, although I agree that the temporal profile of each mutation argues for their different cell of origin.

3) "The presence of a mutation in a DNA damage response gene was associated with a higher total number of variants." TP53 mutations are significantly a fewer numbers of drivers mutations, while are with a higher number of chromosomal abnormalities.

4) What is the putative mechanism of clonal selection of SRCAP mutated cells. Were these mutations loss-of-function? If it is required to DNA damage repair, how SRCAP-mutated cells could survive and be selected?

Reviewer #2:

Remarks to the Author:

The authors have satisfactorily addressed prior comments.

Reviewer #3:

Remarks to the Author:

The revised manuscript entitled, "Cellular stressors contribute to the expansion of hematopoietic clones of varying leukemic potential," is present for review. The authors have addressed each of the reviewer comments. In addition to clarifying several points in the text, the authors have expanded their analysis to include indels in a more comprehensive manner and examined additional clinical associations. The discussion has also been revised in light of reviewer comments.

I have no further suggestions for revision.

However, I am still unclear on how mutations in DNA repair genes are necessarily markers of prior genotoxic stress. My interpretation is that these clones have already adapted to survive with DNA damage repair defects, regardless of prior exposures. The DNA repair defects results in a higher background mutation rate, but it is the adaption to DNA repair defects allows these clones to preferentially survive subsequent cytotoxic insults. Maybe the authors are saying something similar. Further clarification would be useful.

Reviewers' comments:

Reviewer #1 (Remarks to the Author):

In their revised manuscript, authors have addressed majority of the concerns raised by this and other reviewers. Despite authors' rebuts, the major findings in this manuscript have previously been suggested or easily anticipated. Nevertheless, this reviewer believes that it still merits the community, given the growing attention to age-related clonal hematopoiesis of recent years. There are a few additional minor points, in response to this revision.

1) Authors did not include conditioning regimens as "prior cytotoxic therapy exposure" since the time between such exposure and sample collection was typically only a few days. However, usually, it was not only a few days. It should be after a recovery from transient bone marrow suppression, which should surely not be just a few days. During conditioning, it could be possible that clonal selection does occur, provably promoting outgrowth of chemo resistant clones, which might be worth interrogating.

In our cohort, only a single patient who received chemo-mobilization did not receive prior cytotoxic therapy. Reclassifying this patient (who had no mutations) has no impact on the results. In our prior cytotoxic therapy cohort, 16 patients received chemo-mobilization, and 65 patients were mobilized without chemotherapy. Between these two groups, we observed no difference in either the total number mutations or the number of mutations in DNA damage response genes. Since the number of patients is small, we did not include this analysis in the manuscript. However, we modified the discussion to include this as a question for future investigation.

2) "Single-colony" sequencing would be accepted as a widely-used, reliable technique to address the origin of mutations in different population in well-controlled experiments, although I agree that the temporal profile of each mutation argues for their different cell of origin.

We agree that single-cell approaches would be ideal to both assess the heterogeneity of stress-induced hematopoietic clonal expansion and investigate the nature (e.g. presence of additional somatic mutations and/or cytogenetic abnormalities) of expanded hematopoietic clones. Unfortunately, given the low VAFs of our identified variants, our sample set is not the ideal one for this purpose.

3) "The presence of a mutation in a DNA damage response gene was associated with a higher total number of variants." TP53 mutations are significantly a fewer numbers of drivers mutations, while are with a higher number of chromosomal abnormalities.

Our data are most consistent with the expansion of multiple clones, each carrying their own variants, rather than a single clone containing multiple variants. Our discussion of this topic was not clear (see reviewer 3 comments). We apologize for this confusion and have modified the discussion accordingly.

4) What is the putative mechanism of clonal selection of SRCAP mutated cells. Were these mutations loss-of-function? If it is required to DNA damage repair, how SRCAP-mutated cells could survive and be selected?

Of the 29 SRCAP variants identified in pheresis and/or post-transplant peripheral blood samples, 12 (41.4%) were truncating nonsense or frameshift mutations, 16 (55.2%) were

missense mutations, and 1 (3.4%) was a silent mutation. Mutations were scattered throughout the gene with 4 of 16 missense mutation in the catalytic Snf2 domain. This pattern suggests that these are primarily loss-of-function mutations. A brief discussion of these data has been added to the manuscript. The mechanism(s) through which these mutations provide hematopoietic cells with a fitness advantage after cytotoxic therapy exposure is unknown and an area of active investigation.

Reviewer #2 (Remarks to the Author):

The authors have satisfactorily addressed prior comments.

We thank reviewer #2 for the insightful comments in the previous review.

Reviewer #3 (Remarks to the Author):

The revised manuscript entitled, "Cellular stressors contribute to the expansion of hematopoietic clones of varying leukemic potential," is present for review. The authors have addressed each of the reviewer comments. In addition to clarifying several points in the text, the authors have expanded their analysis to include indels in a more comprehensive manner and examined additional clinical associations. The discussion has also been revised in light of reviewer comments.

I have no further suggestions for revision.

We thank reviewer #3 for the insightful comments in the previous review.

However, I am still unclear on how mutations in DNA repair genes are necessarily markers of prior genotoxic stress. My interpretation is that these clones have already adapted to survive with DNA damage repair defects, regardless of prior exposures. The DNA repair defects results in a higher background mutation rate, but it is the adaption to DNA repair defects allows these clones to preferentially survive subsequent cytotoxic insults. Maybe the authors are saying something similar. Further clarification would be useful.

What we meant to say is that the expansion of clones harboring mutations in DNA damage response genes (and not the mutations themselves) may be a marker of prior genotoxic stress. The mutations themselves are present prior to the stress but below our assay's limit of detection. We realize that our discussion on this subject was not entirely clear and have modified it accordingly. We apologize for the confusion.

Reviewers' Comments:

Reviewer #1:

Remarks to the Author:

The authors addressed all the questions and concerns raised by this and other reviewers. I have no further concerns.

We thank Reviewer #1, Reviewer #2, and Reviewer #3 for their insightful comments during the manuscript revision process.

Reviewer's Comments

Reviewer #1 (Remarks to the Author):

The authors addressed all the questions and concerns raised by this and other reviewers. I have no further concerns.